# Synthesis and Molecular Docking Study of Novel Pyrimidine Derivatives against COVID-19

**DOI:** 10.3390/molecules28020739

**Published:** 2023-01-11

**Authors:** Zahra M. Alamshany, Reham R. Khattab, Nasser A. Hassan, Ahmed A. El-Sayed, Mohamed A. Tantawy, Ahmed Mostafa, Allam A. Hassan

**Affiliations:** 1Chemistry Department, Faculty of Science, King Abdulaziz University, Jeddah 21551, Saudi Arabia; 2Photochemistry Department (Synthetic Unit), National Research Centre, Dokki, Giza 12622, Egypt; 3Hormones Department, National Research Centre, Dokki, Giza 12622, Egypt; 4Center of Scientific Excellence for Influenza Viruses, National Research Centre, Dokki, Giza 12622, Egypt; 5Chemistry Department, Faculty of Science, Suez University, Suez 43221, Egypt

**Keywords:** pyridopyrimidine derivatives, phosgeniminium chloride, molecular docking, COVID-19, spectroscopic data

## Abstract

A novel series of pyrido[2,3-*d*]pyrimidines; pyrido[3,2-*e*][1,3,4]triazolo; and tetrazolo[1,5-*c*]pyrimidines were synthesized via different chemical transformations starting from pyrazolo[3,4-*b*]pyridin-6-yl)-*N*,*N*-dimethylcarbamimidic chloride **3b** (prepared from the reaction of *o*-aminonitrile **1b** and phosogen iminiumchloride). The structures of the newly synthesized compounds were elucidated based on spectroscopic data and elemental analyses. Designated compounds are subjected for molecular docking by using Auto Dock Vina software in order to evaluate the antiviral potency for the synthesized compounds against SARS-CoV-2 (2019-nCoV) main protease M ^pro^. The antiviral activity against SARS-CoV-2 showed that tested compounds **7c**, **7d**, and **7e** had the most promising antiviral activity with lower IC_50_ values compared to Lopinavir, “the commonly used protease inhibitor”. Both in silico and in vitro results are in agreement.

## 1. Introduction

Human health is one of the most important factors impacting economic development in the world. So, studying diseases and their potential treatment is an essential research area. The coronavirus disease 2019 (COVID-19) has imposed a great threat globally due its rapid spreading and mutation. Virus-encoded proteases are observed as potential drug targets. The main protease (Mpro) of SRAS-CoV-2 plays a crucial role in the viral maturation.

Studies have reported that the inhibition of Mpro can prevent the virus from replication [1,2,3,4,5]. At present, some medications and vaccines have been approved by the FDA for the treatment of COVID-19 but they could not completely eradicate the virus. The search for safe and effective antivirals today is urgent.

Several synthetic molecules containing heterocyclic moieties are currently known for their antiviral potentials [6,7,8,9,10,11]. Pyrazolo[3,4-*b*]pyridine nucleus is an important scaffold in a variety of marketed anxiolytic antidepressant drugs such as cartazolate, tracazolate and etazolate (Figure 1).

Pyrazolopyrido[2,3-*d*]pyrimidine, a fused hetero-tricyclic nucleus containing pyrazole, pyridine and pyrimidine rings, has attained momentary attention in the sphere of multicomponent synthetic protocol. Pyrazolo[3,4-*b*]pyridine nucleus is an important scaffold for synthesis of lots of bioactive compounds [12,13,14,15,16,17,18,19,20] (Figure 2). Pyrido[2,3-*d*]pyrimidine-derived drugs have manifested diverse pharmacological activities, particularly, anti-inflammatory, cytotoxic, antimicrobial, phosphodiesterase inhibitors and cytokine inhibitors [16,17,18,19,20,21,22].

These pyridine/pyrimidine core structures have been noted for their roles in many biological processes as well as in cancer pathogenesis, which make such compounds attractive scaffolds for discovery of novel drugs. In addition, applications of these derivatives have been found in various areas of medicine, such as anticancer, CNS, fungicidal, antiviral and antibacterial therapies [23,24].

The diamine analogs including pyrido[3,4-*d*]pyrimidine core were reported as tyrosine kinase inhibitors [25,26]. The combination of arylidene hydrazinyl moiety with pyrido[2,3-*d*]pyrimidin-4-one resulted in the output of unprecedented anti-microbial agents [27]. Pyrazolo[4′,3′:5,6]pyrido[2,3-*d*]pyrimidines analogs were reported to have diverse pharmacological activities, including as anticonvulsants, antiproliferative agents, anti-inflammatories and analgesic agents, antibacterial, antifungal, antimicrobial and antitumor activities [28,29,30,31]. Further, pyrazolo[4′,3′:5,6]pyrido[3,2-*e*][1,2,4]triazolo[4,3-*a*]pyrimidinone analogs were reported as antimicrobial agents [32].

For the abovementioned benefits and as part of our program investigating syntheses of heterocyclic compounds which have biological significance [33,34,35,36,37,38,39,40,41,42], herein, our focus will be on the synthesis, molecular docking and pharmacological evaluation of novel series of pyrazolo[4′,3′:5,6]pyrido[2,3-*d*]pyrimidines and pyrazolo[4′,3′:5,6]pyrido[3,2-*e*][1,2,4]triazolo[4,3-*a*]pyrimidinone, which could serve as a promising scaffold in the development of novel viral protease inhibitors of SARS-CoV-2.

## 2. Results and Discussions

### 2.1. Chemistry

The requisite starting material 6-amino-pyrazolo[3,4-b]pyridine-5-carbonitrile **1** was used in this study. It was synthesized according to the known procedure [33] from pyrazolone, arylaldehyde and malononitrile (Figure 1). Phosgene iminium chloride (PIC = dichloromethylene-dimethyliminium chloride) is an efficient reagent in synthetic chemistry and was used to introduce the amide chloride group into activated substrates. So, stirring of compound **1a**,**b** with phosgenimonium chloride in 1,2-dichloroethane at room temperature provided pyridopyrimidine derivative **2a**,**b** and Pyrazolopyridinyl derivative **4b**, respectively, through the pathway outlined in Scheme **1**.

The structures of **2a**,**b** and **4b** were confirmed by elemental analysis and spectral (MS, IR and ^1^H NMR) data (Appendix A) (c.f. experimental). Their ^1^H NMR spectrum in DMSO revealed in each case a characteristic signal in the region of δ 3.10–3.27 assignable to the –N(CH_3_)_2_ proton. Their IR spectra showed the absence of the characteristic band for nitrile group for compounds **2a**,**b** which is present for compound **4b** at 2217 cm^−1^. Additionally, a new peak at ν 735 cm^−1^ for the new C-Cl appeared. Stirring of the compounds **2a**,**b** and **4b** with hydrazine hydrate in ethanol afforded the corresponding 5-imino-pyrazolo[4′,3′:5,6]pyrido[2,3-*d*]pyrimidine derivatives **3a**,**b** and 5-hydrazineyl-pyrazolo[4′,3′:5,6]pyrido[2,3-*d*]pyrimidine derivative **6b**, respectively (Figure 1). Both spectroscopic data and elemental analyses were consistent with the assigned structures (c.f., experimental). IR spectra of compounds **6b** as an example indicated the disappearance the absorption band characteristic for CN and C-Cl groups with the appearance of peaks corresponding to -NH_2_ and NH groups. In the ^1^H NMR spectrum (Appendix A) and in the presence of D_2_O, the signals due to the NH and NH_2_ groups in each compound disappeared and the signals related to methyl, aryl, and N(CH_3_)_2_ protons appeared at the expected values.

Refluxing of hydrazine derivative **3a** with aldehydes and/or monosaccharaides in ethanol, with adding a few drops of acetic acid as a catalyst, gave the corresponding hydrazones derivatives **7c–g**, respectively (Figure 2). The structures of the latter products were confirmed by elemental and spectral analyses. The IR spectrum of the subsequent sugar hydrazones **7f,g** as an example indicated the characteristic bands for the hydroxyl groups at (3415–3419) cm^−1^ in the sugar chain. The ^1^H NMR (c.f. experimental) spectra of compound **7g,h** demonstrated the signs of H-1 methine proton showing up as doublet at δ 7.17 and 7.20 ppm, respectively. Their ^13^C NMR spectra (Appendix A) of compound **7f** revealed five resonances for sugar carbon atoms at δ 61.63, 70.59, 73.75, 75.48, 77.22 ppm. Treatment of each of the hydrazones **7c,d** with iron (III) chloride in ethanol or with potassium iodide in DMSO gave, in each case, a single product as evidenced by TLC analysis. Elemental analyses and mass spectra revealed that each of such isolated products has two hydrogens less than the respective hydrazone. This finding was confirmed by the ^1^H NMR spectra, which indicated the absence of the –N=CH– and hydrazone–NH-N=C protons. Based on this finding, the isolated products were assigned the structure **8c,d** (Figure 2). The conversion of **7** into **8** is reminiscent of other related oxidative cyclization of aldehyde *N*-heteroarylhydrazones with iron(III) chloride, which have been reported to proceed via generation of the respective nitrilimines, which undergo in situ 1,5-electrocyclization to give the respective fused heterocycles [43,44]. Similarly, refluxing of amino derivative **6b** with 4-chlorobenzaldehyde and 4-bromobenzaldehyde and under the same reaction conditions afforded the products **10c,d** via **9c,d** (Figure 3).

When the pyrazolo[4,3-*e*][1,2,4]triazolo[4,3-*c*]pyrimidine derivative **6b** was heated in ethanol in the presence of sodium acetate, it gave a product which proved identical in all respects (mp, mixed mp, IR and ^1^H NMR spectra) with **3b**. It has been found that compound **6b** isomerized to the thermodynamically more stable pyrazolo[4,3-*e*][1,2,4] [1,5-*c*]pyrimidine derivative **3b** through tandem ring opening and ring closure reactions (Figure 4). This rearrangement is consistent with those reported in some earlier reports [45,46].

Compound **3a** was utilized for synthesis of several new compounds. So, product **11** was obtained by stirring **3a** with sodium nitrite in acetic acid. Compound **3a** reacted with carbon disulfide to yield triazolopyrimidinethione **12**. Compound **3a** was converted into compound **13** by reaction with triethyl orthoformate. Dimethyldichloromethylenirninium chloride condensed with **3a** to yield **14** (Figure 5). Structural assignments of all compounds **11**–**14** were based on their elemental analyses and spectroscopic data (c.f., experimental).

### 2.2. Antiviral Assay

To experimentally evaluate the antiviral activity of the compounds showing promising activity based on the virtual analysis, the cytotoxicity (CC_50_) and antiviral activity (IC_50_) were evaluated (Figure 3 and Table 1). Compound **7c** (IC_50_ = 1.2), **7d** (IC_50_ = 2.34) and 7E (IC_50_ = 2.3) displayed promising antiviral activity against SARS-CoV-2 (Alpha strain, isolate hCoV-19/Egypt/NRC-3/2020 SARS-CoV-2 “NRC-03-nhCoV” virus) [47] with wider selectivity indices (SI = 71–130). Compounds **7f** and **10c** showed a moderate potency and a lower selectivity indices (SI = 11–19) (Figure 3). Interestingly, the compounds **7c–7e** showed better IC_50_ and SI values when compared to the commonly used protease inhibitor [48], namely, Lopinavir (IC_50_ = 5.246, SI = 8.57).

Based on anti-viral assay, we would endorse compound **7c** for further detailed investigation, as it showed the best selectivity index (130), which is even better than the Lopinavir drug.

### 2.3. Molecular Docking

To pick up the mode of action of the tested compounds, a molecular-docking study was utilized to determine the binding modes against SARS-CoV-2 main protease “M^pro^”, which are significant targets to create anti-SARS-CoV2 agents. These targets were chosen dependent on their key roles in viral protein formation; therefore, targeting these proteins give potential advantages in killing the virus. The co-crystalized ligand “{tert}-butyl ~{N}-[1-[(2~{S})-3-cyclopropyl-1-oxidanylidene-1-[[(2~{S},3~{R})-3-oxidanyl-4-oxidanyl-idene-1-[(3~{S})-2-oxidanylidenepyrrolidin-3-yl]-4-[(phenylmethyl)amino]butan-2-yl]-amino]-propan-2-yl]-2-oxidanylidene-pyridin-3-yl]carbamate” was re-docked to guarantee the validity of the docking parameters and methods using Auto Dock vina to represent the position and orientation of the ligand recognized in the crystal structure. The distinction of RMSD value between co-crystalized ligands to the original co-crystal ligand was <2 Å, which affirmed the accuracy of the docking protocols and parameters. As reference ligand, the co-crystalized ligand interacted with the active site of the M^pro^ protein via 5 hydrogen bonds, with the active amino acid residues (THR26, GLU166, CSY145, CYS145, SER144) with bond distance (3.139, 3.299, 2.890, 2.959, 3.023, respectively), and the value of the free binding energy was –7.5 Kcal/mol (Table 2 and Table 3) (Figure 3). Comparing our tested compounds based on their binding to M^pro^ protein, 6 compounds (7c, 7d, 7e, 7f, 10c, and 10d) showed preferable binding affinity to the M^pro^ protein more than the co-crystalized ligand did, which was evidenced by the lower values of their free binding energy (−8.4, −8.3, −8.5, −8.0, −8.1, and −8.1 Kcal/mol, respectively), and by the hydrogen bond interaction with the key amino acid residues in the active site of the M^pro^ protein (Table 2 and Table 3) (Figure 4). Based on the docking results, we have selected the most promising tested compounds, and ordered them according to their activities (7e > 7c > 7d > 7f >10c = 10d). Therefore, these compounds may have interesting applications against the Alpha variant of SARS-CoV-2. It is encouraging to expand this series via the synthesis of more analogues in an optically pure form, and to test them all experimentally.

## 3. Structure Activity Relationships

The structure-based activity analysis (SAR) of synthesized compounds revealed the compounds having an electron-withdrawing group bonded to the phenyl ring as shown in **7c** (IC_50_ = 1.2), **7d** (IC_50_ = 2.34), and an electron-donating group bonded to the phenyl ring as observed in **7e** (IC_50_ = 2.3), displaying promising antiviral activity against SARS-CoV-2 with wider selectivity indices (SI = 71–130). The presence of a sugar moiety in compounds **7f**, **7g**, **10f**, and **10g** did not result in increased antiviral activity. Compound **10c** showed moderate activity.

## 4. Materials and Methods

### 4.1. General

All melting points were measured on a Gallenkamp Melting point apparatus and are uncorrected. The IR spectra were recorded on a Shimadzu FT-IR 8101 PC infrared spectrophotometer (Shimadzu, Tokyo, Japan) using KBr disks. The NMR spectra were preserved on a Varian Mercury VX-400 NMR spectrometer (Varian, Palo Alto, CA, USA). ^1^H NMR spectra were run at 400 MHz and ^13^C NMR spectra were run at 75.46 MHz in deuterated chloroform (CDCl_3_) or dimethyl sulfoxide (DMSO-*d_6_*) as specified in individual compound characterizations. Chemical shifts are given in parts per million and were referenced to those of the solvents. Mass spectra were recorded on a Shimadzu GCMS-QP 1000 EX mass spectrometer at 70 eV. Elemental analyses were registered on an Elementar-Vario EL (Germany) automatic analyzer.

### 4.2. Synthetic Procedures

#### 4.2.1. General Procedure for Synthesis of Starting 6-Amino-3-Methyl-4-Aryl-Pyrazolo[3,4-b]Pyridine-5-Carbonitrile **1a,b**

A solution of pyrazolone **1** (0.98 g, 0.01 mol), arylaldehyde (0.01 mol), and malononitrile (0.66 g, 0.01 mol) in ethanol (10 mL) containing Ammonium acetate (0.98 gm, 2% excess) or piperidine (1 mL) was heated under reflux for 5 h. The solvent was evaporated under vacuum and the remaining solids were treated with the proper solvent for crystallization. The spectroscopic data and melting points of **1a,b** agreed with those reported [33].

#### 4.2.2. General Procedure for Synthesis of Compounds **2a**, **2b**, and **4b**

A solution of **1a, b** (10 mmol) and phosogen iminiumchloride (11 mmol) in dry 1,2-dichloroethane (100 mL) was stirred at the temperature and time mentioned in the Figure 1. The precipitated solid was collected by filtration and recrystallized from dioxane to afford the title products.

5-Chloro-*N*,*N*,3-trimethyl-4-(4-nitrophenyl)-4,9-dihydro-1*H*-pyrazolo[4′,3′:5,6]pyrido-[2,3-*d*]pyrimidin-7-amine (**2a**)

Yield (65%) as yellow powder; mp 213-215 °C. IR (KBr, υ_max_, cm^−1^): 3428 (NH)_,_ 2927 (CH alkyl). ^1^H NMR (DMSO-*d_6_*) δ_H_ ppm: 2.09 (s, 3H, CH_3_), 3.23 (s, 6H, N-Me_2_), 5.50 (s, 1H, CH), 7.53–7.55 (m, 3H, Ar-H and NH), 8.16–8.18 (d, 2H, *J* = 8.6 Hz, Ar-H), 12.05 (s, 1H, NH, exchangeable with D_2_O). ^13^C NMR (DMSO-*d_6_*) δ_C_ ppm: 14.6, 32.0, 35.3, 94.8, 103.9, 105.8, 111.9, 116.5, 121.7, 127.6, 132.9, 139.3, 144.6, 147.6, 158.3, 161.3, 163.3. (*m/z*, %) (386, 10). Anal. Calcd. for C_17_H_16_ClN_7_O_2_ (385.81): C, 52.92; H, 4.18; Cl, 9.19; N, 25.41;. Found: C, 52.99; H, 4.27; Cl, 9.12; N, 25.32%.

5-Chloro-*N*,*N*,3-trimethyl-4-(thiophen-2-yl)-4,9-dihydro-1*H*-pyrazolo[4′,3′:5,6]pyrido-[2,3-*d*]pyrimidin-7-amine (**2b**)

Yield (69%) as yellow powder; mp 228–232 °C. IR (KBr, υ_max_, cm^−1^): 3432 (NH)_,_ 2935 (CH alkyl). ^1^H NMR (DMSO-*d_6_*) δ_H_ ppm: 2.11 (s, 3H, CH_3_), 3.16 (s, 6H, N-Me_2_), 5.48 (s, 1H, CH), 6.89 (m, 3H, thiophene-H and NH), 7.25 (d, 1H, *J* = 4.9 Hz, thiophene-H), 11.90 (s, 1H, NH, exchangeable with D_2_O); MS (*m/z*, %) (346, 48). Anal. Calcd. for C_15_H_15_ClN_6_S (346.84): C, 51.95; H, 4.36; Cl, 10.22; N, 24.23; S, 9.24. Found: C, 51.84; H, 4.45; Cl, 10.27; N, 24.19; S, 9.28%.

*N*′-(5-Cyano-3-methyl-4-(thiophen-2-yl)-4,7-dihydro-1*H*-pyrazolo[3,4-*b*]pyridin-6-yl)-*N*,*N*-dimethylcarbamimidic chloride (**4b**)

Yield: 60%, mp 151–152 °C. IR (KBr, υ_max_, cm^−1^): 3440 (NH), 2212 (CN), 2998 (CH alkyl). ^1^H NMR (DMSO-*d6*) δ_H_ ppm: 2.14 (s, 3H, CH_3_), 3.15 (s, 6H, N-Me_2_), 5.46 (s, 1H,CH), 6.89 (m, 3H, thiophene-H, NH), 7.25 (d,1H, *J* = 4.9 Hz, thiophene-H),; 11.90 (br s, 1H, NH, exchangeable with D_2_O). MS (*m/z*, %) (346, 48). Anal. Calcd. for C_15_H_15_ClN_6_S (346.84): C, 51.95; H, 4.36; Cl, 10.22; N, 24.23; S, 9.24. Found: C, 51.84; H, 4.45; Cl, 10.27; N, 24.19; S, 9.28%.

#### 4.2.3. Synthesis of **3a**, **3b**, **6b**

A solution of **2a**, **b,** and **4b** (10 mmol) and hydrazine hydrate (11 mmol) in EtOH (40 mL) were stirring for 10 h at room temperature. The product**s 3a**, **3b,** and **6b** were collected by filtration and recrystallized from dioxane**.**

5-Hydrazineyl-*N*,*N*,3-trimethyl-4-(4-nitrophenyl)-4,9-dihydro-1*H*-pyrazolo[4′,3′:5,6]-pyrido[2,3-*d*]pyrimidin-7-amine (**3a**)

Yield (61%) as brown powder; mp 230–232 °C. IR (KBr, υ_max_, cm^−1^): 3430 (NH + NH_2_)_,_ 2923 (CH alkyl). ^1^H NMR (DMSO-*d_6_*) δ_H_ ppm: 1.96 (s, 3H, CH_3_), 3.09 (s, 6H, N-Me_2_), 4.36 (br, 2H, NH_2_, exchangeable with D_2_O), 5.26 (s, 1H, CH), 7.56–7.67 (m, 2H, Ar-H and NH), 7.70–8.1 (d, 2H, *J* = 8.6 Hz, Ar-H), 8.18 (m, 2H, Ar-H), 12.06 (s, 1H, NH, exchangeable with D_2_O). ^13^C NMR (DMSO-*d_6_*) δ_C_ ppm: 21.0, 29.1, 30.4, 98.6, 99.3, 109.8, 115.8, 124.8, 128.7, 129.4, 135.4, 139.9, 143.4, 149.0, 158.1, 160.5, 166.8. MS (*m/z*, %) (381, 60). Anal. Calcd. for C_17_H_19_N_9_O_2_ (381.40): C, 53.54; H, 5.02; N, 33.05. Found. C, 53.60; H, 5.08; N, 33.10%.

5-Hydrazineyl-*N*,*N*,3-trimethyl-4-(thiophen-2-yl)-4,9-dihydro-1*H*-pyrazolo[4′,3′:5,6]-pyrido[2,3-*d*]pyrimidin-7-amine (**3b**)

Yield (65%) as brown powder; mp 236–238 °C. IR (KBr, υ_max_, cm^−1^): 3436 (NH + NH_2_)_,_ 2929 (CH alkyl). ^1^H NMR (DMSO-*d6*) δ_H_ ppm: 2.12 (s, 3H, CH_3_), 3.39 (s, 6H, N-Me_2_), 5.37 (s, 1H, CH), 5.45 (br s, 2H, NH_2_, exchangeable with D_2_O), 6.76–6.83 (m, 3H, thiophene-H and NH), 7.23 (d, 1H, *J* = 4.9 Hz, thiophene-H), 8.83 (br s, 1H, NH, exchangeable with D_2_O), 12.11 (br s, 1H, NH, exchangeable with D_2_O). MS (*m/z*, %) (342, 68). Anal. Calcd. for C_15_H_18_N_8_S (342.43): C, 52.61; H, 5.30; N, 32.72; S, 9.36. Found. C, 52.52; H, 5.26; N, 32.84; S, 9.39%.

5-Imino-*N*,*N*,3-trimethyl-4-(thiophen-2-yl)-1,4,5,9-tetrahydro-6*H*-pyrazolo[4′,3′:5,6]-pyrido[2,3-*d*]pyrimidine-6,7-diamine (**6b**)

Yield (60%) as brown powder; mp 239–241 °C. IR (KBr, υ_max_, cm^−1^): 3425 (NH + NH_2_)_,_ 2925 (CH alkyl). ^1^H NMR (DMSO-*d_6_*) δ_H_ ppm: 2.13 (s, 3H, CH_3_), 3.35 (s, 6H, N-Me_2_), 4.73 (s, 2H, NH_2_, exchangeable with D_2_O), 5.50 (s, 1H, CH), 6.99–7.11 (m, 3H, thiophene-H and NH), 7.35(d, 1H, *J* = 4.9 Hz, thiophene-H), 11.28 (s, 1H, NH, exchangeable with D_2_O); 12.34 (s, 1H, NH, exchangeable with D_2_O). ^13^C NMR (DMSO-*d_6_*) δ_C_ ppm: 14.8, 27.7, 30.1, 89.6, 99.0, 115.8, 124.8, 129.7, 131.8, 139.2, 139.6, 143.4, 157.8, 159.8, 160.5. MS (*m/z*, %) (342, 63). Anal. Calcd. for C_15_H_18_N_8_S (342.43): C, 52.61; H, 5.30; N, 32.72; S, 9.36. Found. C, 52.43; H, 5.10; N, 32.92; S, 9.46%.

#### 4.2.4. General Procedure for the Synthesis of **7c–g** and **9c,d**

To a mixture of 1 (0.6 g, 2.5 mmol) and the appropriate aldose 2a–d (2.5 mmol) in ethanol (15 ml), a catalytic amount of glacial acetic acid (0.1 ml) was added

To a mixture of derivatives **3a** or **6b** (10 mmol) and the appropriate aldehyde (10 mmol) or respective monosaccharides (10 mmol) in ethanol (30 mL), a catalytic amount of glacial acetic acid (1.0 mL) was added. The reaction mixture was refluxed for eight hours. After cooling at room temperature, the precipitated solid was collected by filtration and recrystallised from the proper solvent.

5-(2-(4-Chlorobenzylidene)hydrazineyl)-*N*,*N*,3-trimethyl-4-(4-nitrophenyl)-4,9-dihydro-1*H*-pyrazolo[4′,3′:5,6]pyrido[2,3-*d*]pyrimidin-7-amine (**7c**)

Yield (70%) as brown powder; mp 189–191 °C. IR (KBr, υ_max_, cm^−1^): 3417 (NH), 2930 (CH alkyl). ^1^H NMR (DMSO-*d6*) δ_H_ ppm: 2.17 (s, 3H, CH_3_), 3.11 (s, 6H, N-Me_2_), 6.08 (s, 1H, CH), 6.85–6.89 (m, 3H, Ar-H and NH), 7.22–7.42 (d, 2H, *J* = 8.4 Hz, Ar-H), 7.50–7.52 (m, 2H, *J* = 8.4Hz, Ar-H), 7.68-7.80 (d, 2H, *J* = 8.4 Hz, Ar-H), 8.12 (s, 1H, N=CH), 10.52 (s, 1H, NH, exchangeable with D_2_O), 12.12 (s, 1H, NH, exchangeable with D_2_O). ^13^C NMR (DMSO-*d_6_*) δ_C_ ppm: 21.1, 30.1, 30.4, 89.9, 99.3, 115.8, 119.24, 119.9, 120.3, 120.7, 121.4, 124.5, 129.1, 132.9, 134.9, 136.7, 138.1, 139.9, 143.7, 147.9, 157.8, 159.8, 166.5, 167.8. MS (*m/z*, %) (503, 53). Anal. Calcd. for C_24_H_22_ClN_9_O_2_ (503.95): C, 57.20; H, 4.40; Cl, 7.03; N, 25.01. Found. C, 57.38; H, 4.61; Cl, 6.88; N, 24.80%.

5-(2-(4-Bromobenzylidene)hydrazineyl)-*N*,*N*,3-trimethyl-4-(4-nitrophenyl)-4,9-dihydro-1*H*-pyrazolo[4′,3′:5,6]pyrido[2,3-*d*]pyrimidin-7-amine (**7d**)

Yield (67%) as brown powder; mp 180–182 °C. IR (KBr, υ_max_, cm^−1^): 3422 (NH), 2940 (CH alkyl). ^1^H NMR (DMSO-*d_6_*) δ_H_ ppm: 2.13 (s, 3H, (CH_3_), 3.34 (s, 6H, N-Me_2_), 5.85 (s, 1H, CH), 7.43–7.53 (m, 3H, Ar-H and NH), 7.55–7.60 (d, 2H, *J* = 8.4 Hz, Ar-H), 7.67 (m, 2H, *J* = 8.4Hz, Ar-H), 7.92 (d, 2H, *J* = 8.4 Hz, Ar-H), 8.07 (s, 1H, N=CH), 10.63 (s, 1H, NH, exchangeable with D_2_O), 12.19 (s, 1H, NH, exchangeable with D_2_O). MS (*m/z*, %) (548, 71). Anal. Calcd. for C_24_H_22_BrN_9_O_2_ (548.41): C, 52.56; H, 4.04; Br, 14.57; N, 22.99. Found. C, 52.39; H, 4.29; Br, 14.31; N, 23.12%.

(E)-*N*,N,3-Trimethyl-5-(2-(4-methylbenzylidene)hydrazineyl)-4-(4-nitrophenyl)-4,9-dihydro-1*H*-pyrazolo[4′,3′:5,6]pyrido[2,3-*d*]pyrimidin-7-amine (**7e**)

Yield (66%) as brown powder; mp 188–190 °C. IR (KBr, υ_max_, cm^−1^): 3420 (NH), 2945 (CH alkyl). ^1^H NMR (DMSO-*d_6_*) δ_H_ ppm: 2.17 (s, 3H, CH_3_), 2.43 (s, 3H, CH_3_), 3.37 (s, 6H, N-Me_2_), 5.91 (s, 1H, CH), 7.31–7.35 (m, 3H, Ar-H and NH), 7.49 (d, 2H, *J* = 8.4 Hz, Ar-H), 7.71 (d, 2H, *J* = 8.4 Hz, Ar-H), 7.95 (d, 2H, *J* = 8.4 Hz, Ar-H), 8.09 (s, 1H, N=CH), 10.56 (s, 1H, NH, exchangeable with D_2_O), 12.06 (s, 1H, NH, exchangeable with D_2_O). MS (*m/z*, %) (483, 66). Anal. Calcd. for C_25_H_25_N_9_O (483.54): C, 62.10; H, 5.21; N, 26.07; O, 6.62. Found. C, 62.29; H, 5.08; N, 26.27%.

5-(2-(4-Glucosylhydrazinyl)hydrazineyl)-*N*,*N*,3-trimethyl-4-(4-nitrophenyl)-4,9-dihydro-1*H*-pyrazolo[4′,3′:5,6]pyrido[2,3-*d*]pyrimidin-7-amine (**7f**).

Yield (75%) as brown powder; mp 173–175 °C. IR (KBr, υ_max_, cm^−1^): 3419-3347 (OH + NH). ^1^H NMR (DMSO-*d_6_*) δ_H_ ppm: 2.13 (s, 3H, CH_3_), 3.10 (s, 6H, N-Me_2_), 3.37 (m, 2H, H-6′ and H-6′′), 3.41 (m, 1H, H-5′), 4.27–4.29 (d, 1H, H-4′), 4.44–4.51 (m, 2H, H-3′ and H-2′), 4.85–4.92 (m, 1H, OH), 5.53 (m, 1H, OH), 5.55 (s, 1H, CH), 5.90 (m, 1H, OH), 6.21 (m, 1H, OH), 6.43 (br s, 1H, NH, exchangeable with D_2_O), 6.59 (m, 1H, OH), 6.99–7.00 (d, 1H, *J* = 7.6 Hz, H-1), 7.10–7.12 (m, 3H, Ar-H and NH), 7.32–7.36 (d, 2H, *J* = 8.4 Hz Ar-H), 12.29 (s, 1H, NH, exchangeable with D_2_O). ^13^C NMR (DMSO-*d_6_*) δ_C_ ppm: 10.2, 34.7, 36.9, 37.1, 61.6, 70.7, 72.4, 75.5, 77.2, 92.7, 97.3, 99.6, 101.9, 124.7, 124.9, 125.0, 127.1, 136.8, 149.5, 150.8, 159.8, 162.1, 164.9. Anal. Calcd. for C_23_H_29_N_9_O_7_ (543.54): C, 50.82; H, 5.38; N, 23.19. Found. C, 50.20; H, 5.51; N, 23.10%.

5-(2-(4-Xylosylhydrazinyl)hydrazineyl)-*N*,*N*,3-trimethyl-4-(4-nitrophenyl)-4,9-dihydro-1*H*-pyrazolo[4′,3′:5,6]pyrido[2,3-*d*]pyrimidin-7-amine (**7g**)

Yield (72%) as brown powder; mp 165–167 °C. IR (KBr, υ_max_, cm^−1^): 3415-3340 (OH + NH). ^1^H NMR (DMSO-*d_6_*) δ_H_ ppm: 2.10 (s, 3H, CH_3_), 3.33 (s, 6H, N-Me_2_), 3.54–4.03 (m, 2H, H-5′ and H-5′′), 4.29–4.45 (m, 2H, H-3′ and H-4′), 4.95 (m, 1H, H-2′), 5.13 (m, 1H, OH), 5.54 (s, 1H, CH), 5.85 (m, 1H, OH), 5.94 (m, 1H, OH), 6.10 (s, 1H, NH, exchangeable with D_2_O), 6.25 (m, 1H, OH), 7.11 (d, 1H, *J* = 7.6 Hz, H-1′), 7.34–7.39 (m, 3H, Ar-CH and NH), 7.59–7.71 (d, 2H, *J* = 8.4 Hz, Ar-CH), 11.99 (br s, 1H, NH, exchangeable with D_2_O); ^13^C NMR (DMSO-*d_6_*) δ_C_ ppm: 14.6, 29.3, 36.6, 37.3, 66.1, 70.1, 75.1, 77.1, 93.2, 98.5, 100.5, 102.5, 124.6, 125.3, 129.9, 134.9, 139.2, 145.9, 152.3, 161.7, 162.7, 165.1. Anal. Calcd. for C_22_H_27_N_9_O_6_ (513.52): C, 51.46; H, 5.30; N, 24.55. Found. C, 51.66; H, 5.10; N, 24.50%.

#### 4.2.5. General Procedure for the Synthesis of **8c,d** and **10c,d**

Method A: Derivatives (7c,d) or (9c,d) (5mmol) were dissolved in DMF (15 mL) and (5 mmol) of potassium iodide; the mixture was stirred and heated at 110 °C for 20 h. The reaction was allowed to cool and the precipitate isolated by filtration and washed with a little methanol and recrystallized from dioxane.

Method B: 2 M solution of iron (III) chloride in ethanol (2 mL) was added dropwise to a boiling solution of (7c,d) or (9c,d) (10 mmol) in ethanol (50 mL). Heating was continued for 20 min and the mixture was then kept overnight at room temperature. The reaction was allowed to cool and the precipitate isolated by filtration and washed with a little methanol and recrystallized from dioxane.

3-(4-Chlorophenyl)-*N*,*N*,10-trimethyl-11-(4-nitrophenyl)-8,11-dihydro-7*H*-pyrazolo-4′,3′:5,6]pyrido[3,2-*e*][1,2,4]triazolo[4,3-*c*]pyrimidin-5-amine (**8c**)

Yield (63%) as brown powder; mp 250–252 °C. IR (KBr, υ_max_, cm^−1^): 3430 (NH). ^1^H NMR (DMSO-*d_6_*) δ_H_ ppm: 2.31 (s, 3H, CH_3_), 3.36 (s, 6H, N-Me_2_), 5.84 (s, 1H, CH), 7.33–7.53 (m, 3H, Ar-CH and NH), 7.74 (d, 2H, *J* = 8.4 Hz, Ar-CH), 8.09 (d, 2H, *J* = 8.4 Hz, Ar-CH), 8.18 (d, 2H, *J* = 8.4 Hz, Ar-CH), 12.03 (s, 1H, NH, exchangeable with D_2_O). Anal. Calcd. for C_24_H_20_ClN_9_O_2_ (501.94): C, 57.43; H, 4.02; Cl, 7.06; N, 25.12. Found: C, 57.60; H, 4.20; Cl, 6.90; N, 25.05%.

3-(4-Bromophenyl)-*N*,*N*,10-trimethyl-11-(4-nitrophenyl)-8,11-dihydro-7*H*-pyrazolo-[4′,3′:5,6]pyrido[3,2-e][1,2,4]triazolo[4,3-*c*]pyrimidin-5-amine (**8d**)

Yield (70%) as brown powder; mp 247–249 °C. IR (KBr, υ_max_, cm^−1^): 3420 (NH). ^1^H NMR (DMSO-*d_6_*) δ_H_ ppm: 2.29 (s, 3H, CH_3_), 3.38 (s, 6H, N-Me_2_), 5.80 (s, 1H,CH), 7.50–7.55 (m, 3H, Ar-CH, NH), 7.70 (d, 2H, *J* = 8.4 Hz, Ar-CH), 8.00 (d, 2H, *J* = 8.4 Hz, Ar-CH), 8.20 (d, 2H, *J* = 8.4 Hz, Ar-CH), 12.20 (s, 1H, NH, exchangeable with D_2_O). Anal. Calcd. for C_24_H_20_BrN_9_O_2_ (546.39): C, 52.76; H, 3.69; Br, 14.62; N, 23.07. Found: C, 52.70; H, 3.59; Br, 14.55; N, 23.30%.

(E)-*6*-((4-Chlorobenzylidene)amino)-5-imino-*N*,*N*,3-trimethyl-4-(thiophen-2-yl)-4,5,6,9-tetrahydro-1*H*-pyrazolo[4′,3′:5,*6*]pyrido[2,3-d]pyrimidin-7-amine (**9c**)

Yield (68%) as brown powder; mp 183–185 °C. IR (KBr, υ_max_, cm^−1^): 3436 (NH), 2936 (CH alkyl). ^1^H NMR (DMSO-*d_6_*) δ_H_ ppm: 2.13 (s, 3H, CH_3_), 3.39 (s, 6H, N-Me_2_), 6.03 (s, 1H, CH), 7.19–7.27 (m, 3H, thiophene-H, NH), 7.47 (d, 1H, *J* = 4.9 Hz, thiophene-H), 7.54 (d, 2H, *J* = 8.4 Hz, Ar-H), 7.64 (d, 2H, *J* = 8.4 Hz, Ar-H), 8.07 (s, 1H, N=CH), 11.05 (s, 1H, NH, exchangeable with D_2_O), 12.12 (s, 1H, NH, exchangeable with D_2_O). MS (*m/z*, %) (464, 59). Anal. Calcd. for C_22_H_21_ClN_8_S (464.98): C, 56.83; H, 4.55; Cl, 7.62; N, 24.10; S, 6.89. Found. C, 56.58; H, 4.78; Cl, 7.47; N, 24.230; S, 6.80%.

(E)-*6*-((4-Bromobenzylidene)amino)-5-imino-*N*,*N*,3-trimethyl-4-(thiophen-2-yl)-4,5,6,9-tetrahydro-1*H*-pyrazolo[4′,3′:5,6]pyrido[2,3-*d*]pyrimidin-7-amine (**9d**)

Yield (60%) as brown powder; mp 199–201 °C. IR (KBr, υ_max_, cm^−1^): 3410 (NH), 2923 (CH alkyl). ^1^H NMR (DMSO-*d_6_*) δ_H_ ppm: 2.18 (s, 3H, CH_3_), 3.40 (s, 6H, N-Me_2_), 6.03 (s, 1H, CH), 6.89–6.92 (m, 3H, thiophene-H and NH), 7.30 (d, 1H, *J* = 4.9 Hz, thiophene-H), 7.58 (d, 2H, *J* = 8.4 Hz, Ar-H), 7.70 (d, 2H, *J* =8.4 Hz, Ar-H), 8.23 (s, 1H, N=CH), 10.50 (s, 1H, NH, exchangeable with D_2_O), 12.00 (s, 1H, NH, exchangeable with D_2_O). MS (*m/z*, %) (509, 41). Anal. Calcd. for C_22_H_21_BrN_8_S (509.43): C, 51.87; H, 4.16; Br, 15.68; N, 22.00; S, 6.29. Found. C, 51.57; H, 4.40; Br, 15.40; N, 22.31; S, 6.21%.

2-(4-Chlorophenyl)-*N*,*N*,10-trimethyl-11-(thiophen-2-yl)-8,11-dihydro-7*H*-pyrazolo-[4′,3′:5,6]pyrido[3,2-*e*][1,2,4]triazolo[1,5-*c*]pyrimidin-5-amine (**10c**)

Yield (60%) as brown powder; mp 219–221 °C. IR (KBr, υ_max_, cm^−1^): 3448 (NH). ^1^H NMR (DMSO-*d_6_*) δ_H_ ppm: 2.13 (s, 3H, CH_3_), 3.41 (s, 6H, N-Me_2_), 5.80 (s, 1H, CH), 6.79–7.01 (m, 3H, thiophene-H and NH), 7.23 (d, 1H, *J = 4.9 Hz*, thiophene-H), 7.51 (d, 2H, *J* = 8.4 Hz, Ar-H), 7.69 (d, 2H, *J* = 8.4 Hz, Ar-H), 11.94 (s, 1H, NH, exchangeable with D_2_O). MS (*m/z*, %) (462, 18). Anal. Calcd. for C_22_H_19_ClN_8_S (462.96): C, 57.08; H, 4.14; Cl, 7.66; N, 24.20; S, 6.93. Found: C, 57.20; H, 4.30; Cl, 7.60; N, 24.06; S, 6.89%.

2-(4-Bromophenyl)-*N*,*N*,10-trimethyl-11-(thiophen-2-yl)-8,11-dihydro-7*H*-pyrazolo-[4′,3′:5,6]pyrido[3,2-*e*][1,2,4]triazolo[1,5-*c*]pyrimidin-5-amine (**10d**)

Yield (67%) as brown powder; mp 222–224 °C. IR (KBr, υ_max_, cm^−1^): 3440 (NH). ^1^H NMR (DMSO-*d6*) δH ppm: 2.12 (s, 3H, CH_3_), 3.47 (s, 6H, N-Me_2_), 5.82 (s, 1H, CH), 6.75-6.80 (m, 3H, thiophene-H and NH), 7.27 (d, 1H, *J* = 4.9 Hz, thiophene-H), 7.59 (d, 2H, *J* = 8.4 Hz, Ar-H), 7.73 (d, 2H, *J* = 8.4 Hz, Ar-H), 11.97 (s, 1H, NH, exchangeable with D_2_O). Anal. Calcd. for C_22_H_19_BrN_8_S (507.41): C, 52.08; H, 3.77; Br, 15.75; N, 22.08; S, 6.32. Found: C, 52.18; H, 3.80; Br, 15.89; N, 21.88; S, 6.11%.

*N,N,*10-Trimethyl-11-(4-nitrophenyl)-8,11-dihydro-7*H*-pyrazolo[4′,3′:5,6]-pyrido[3,2-*e*]tetrazolo[1,5-*c*]pyrimidin-5-amine (**11**)

A solution of **3a** (4.01 g, 10 mmol) and sodium nitrite (0.69 g, mmol) in acetic acid (50 mL) was stirred at room temperature for 24 h. Cold distilled water was added and the precipitate was collected by filtration and crystallized from dioxane. Yield (70%) as yellow powder; mp 225–227 °C; IR ν 3425 cm^−1^ (NH), 2928 cm^−1^ (CH alkyl); ^1^H NMR (DMSO-*d6*) δH ppm: 2.15 (s, 3H, CH_3_), 3.43 (s, 6H, N-Me_2_), 5.52 (s, 1H, CH), 7.66–7.75 (m, 3H, Ar-H and NH), 8.20 (d, 2H, *J* = 8.3 Hz, Ar-H), 12.13 (s, 1H, NH, exchangeable with D_2_O). Anal. Calcd. for C_17_H_16_N_10_O_2_ (392.38): C, 52.04; H, 4.11; N, 35.70. Found. C, 52.21; H, 3.91; N, 35.61%.

5-(Dimethylamino)-10-methyl-11-(4-nitrophenyl)-2,7,8,11-tetrahydro-3*H*-pyrazolo-[4′,3′:5,6]pyrido[3,2-*e*][1,2,4]triazolo[4,3-*c*]pyrimidine-3-thione (**12**)

To a warm ethanolic sodium hydroxide solution (prepared by refluxing (0.40 g, 10 mol) of NaOH in abs. EtOH (50 mL)) was added (10 mmol) of compound **3a** and carbon disulfide (15 mmol). The mixture was heated on a water bath for 6 h, then allowed to cool, poured into water, and neutralized with diluted HCl. The solid product was collected by filtration and crystallized from benzene. Yield (62%) as brown powder; mp 210–212 °C. IR (KBr, υ_max_, cm^−1^): 3430 (NH), 2934 (CH alkyl). ^1^H NMR (DMSO-*d6*) δ_H_ ppm: 2.19 (s, 3H, CH_3_), 3.38 (s, 6H, N-Me_2_), 5.60 (s, 1H, CH), 7.61–7.68 (m, 3H, Ar-H and NH), 8.16 (d, 2H, *J* = 8.3 Hz, Ar-H), 8.23 (s, 1H, -SH), 12.10 (br s, 1H, NH, exchangeable with D_2_O); (M-1, %) (422, 2). Anal. Calcd. for C_18_H_17_N_9_O_2_S (423.46): C, 51.06; H, 4.05; N, 29.77; S, 7.57. Found. C, 51.20; H, 4.19; N, 29.52; S, 7.30%.

*N*,*N*,10-Trimethyl-11-(4-nitrophenyl)-8,11-dihydro-7*H*-pyrazolo[4′,3′:5,6]pyrido[3,2-*e*]-[1,2,4]triazolo[4,3-*c*]pyrimidin-5-amine (**13**)

A solution of **3a** (10 mmol) in triethyl orthoformate (25 mL) was stirred at 70 °C for 9 h and then cooled overnight. The solid product, so-formed, was collected by filtration and crystallized from methanol.Yield (67%) as brown powder; mp 215–217 °C. IR (KBr, υ_max_, cm^−1^): 3408 (NH), 2947 (CH alkyl). ^1^H NMR (DMSO-*d6*) δH ppm: 2.12 (s, 3H, CH_3_), 3.44 (s, 6H, N-Me_2_), 5.61 (s, 1H, CH), 7.57-7.60 (m, 3H, Ar-H and NH), 8.10 (d, 2H, *J* = 8.3 Hz, Ar-H), 8.55 (s, 1H, CH),12.10 (s, 1H, NH, exchangeable with D_2_O). (*m/z*, %) (391, 4). Anal. Calcd. for C_18_H_17_N_9_O_2_ (391.40): C, 55.24; H, 4.38; N, 32.21. Found. C, 55.15; H, 4.20; N, 32.10%.

*N^3^*^,^*N^3^*,*N^5^*,*N^5^*,10-Pentamethyl-11-(4-nitrophenyl)-8,11-dihydro-7*H*-pyrazolo[4′,3′:5,6]-pyrido[3,2-*e*][1,2,4]triazolo[4,3-*c*]pyrimidine-3,5-diamine (**14**)

A solution of **3a** (10 mmol) in 1,2-dichloroethane (10 mL) was added to a stirred suspension of phosgene iminium chloride (10 mmol) in 1,2-dichloroethane (30 mL) at room temperature. The mixture was then refluxed for 4 h. The solid product was collected by filtration, washed with saturated solution of NaHCO_3_, H_2_O, dried and crystallized from ethanol. Yield (60%) as brown powder; mp 203–205 °C. IR (KBr, υ_max_, cm^−1^): 3419 (NH), 2946 (CH alkyl). ^1^H NMR (DMSO-*d6*) δ_H_ ppm: 2.17 (s, 3H, CH_3_), 3.36 (s, 6H, N-Me_2_), 3.38 (s, 6H, N-(CH_3_)_2_), 5.59 (s, 1H,CH), 7.68–7.75 (m,3H, Ar-H,NH), 8.19 (d, 2H, *J* = 8.3 Hz, Ar-H), 12.14 (s, 1H, NH, exchangeable with D_2_O); (*m/z*, %) (434, 9). Anal. Calcd. for C_20_H_22_N_10_O_2_ (434.46): C, 55.29; H, 5.10; N, 32.24. Found. C, 55.179; H, 5.22; N, 32.13%.

### 4.3. Cytotoxic Concentration 50 (CC_50_) and Viral Inhibitory Concentration 50 (IC_50_) Calculation

The assay was performed according to the procedure that was previously described (Feoktistova, M., Geserick, P., Leverkus, M. 2016. Crystal Violet Assay for Determining Viability of Cultured Cells. In Cold Spring Harb Protoc, pdb.prot087379) with minor modifications. Vero E6 cells were seeded into 96-well plates in 100 µL of high glucose Dulbecco’s modified Eagle’s medium (DMEM) containing medium 10% fetal bovine serum (FBS), 100 units/mL penicillin, and 100 µg/mL streptomycin at 37 °C in 5% CO_2_. After 24 h (90–100% confluence monolayer of Vero E6), each compound was diluted into varying concentrations in a separate U-shape 96-well plate (with a range of concentration from 10 µg/mL to 1 ng/mL) using DMEM containing 2% FBS (maintenance medium). A volume of 100 µL of each dilution was transferred into a new U-shape 96-well plate and supplemented with 100 TCID_50_ in 100 µL maintenance medium. In parallel, the wells dedicated for CC_50_ calculation were supplemented with 100 µL maintenance medium without virus. Aliquots of 100 µL of infection media containing 100 TCID_50_ were used as virus control. After 1 h of incubation, 100 µL of each well was transferred to the corresponding wells into the 96-well plates containing Vero E6 cultures. The plates were incubated for 72 h, the cell monolayers were washed with PBS and subjected to cell fixation using 100 µL of 10% formalin for 1 h. Subsequently, the plates are washed well 3 times with 1 × PBS and dried well before staining with 50 µL (0.5%) crystal violet to each well ((0.5 g crystal violet powder (Sigma-Aldrich), 80 mL distilled H_2_O and 20 mL methanol)) for 30 min. The plates were then washed well with rinsed water and air-dried at room temperature for 2 to 24 h. To distain crystal violet, 200 µL methanol was added to each well, and the plate was incubated with its lid on a bench rocker (20 oscillations/minute) for 20 min at room temperature. Finally, the optical density of each well at λ 590 nm (OD590) was measured with a plate reader. The average OD of each dilution without or with virus was compared to control cells and control virus wells to calculate CC_50_ and IC_50_ values using nonlinear regression in GraphPad Prism 5.01.

### 4.4. Molecular Docking Study

The structures of all tested compounds were modeled using the Chemsketch software (http://www.acdlabs.com/resources/freeware/, accessed on 15 December 2021). The structures were optimized and energy minimized using the VEGAZZ software [49]. The optimized compounds were used to perform molecular docking. The three-dimensional structures of the molecular target were obtained from Protein Data Bank (PDB) (www.rcsb.org, accessed on 15 December 2021): SARS-CoV-2 (2019-nCoV) main protease M ^pro^ (PDB: 6Y2F, https://www.rcsb.org/structure/6Y2F, accessed on 15 December 2021). The steps for receptor preparation included the removal of heteroatoms (water and ions), the addition of polar hydrogen, and the assignment of charge. The active sites were defined using grid boxes of appropriate sizes around the bound cocrystal ligands. The docking study was performed using Autodock vina [50] and Chimera for visualization [51].

## Data Availability

Data are contained within the article and the Appendix A.

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
