# Peer review of "Synthesis and Molecular Docking Study of Novel Pyrimidine Derivatives against COVID-19"

_molecules, 2023, doi:10.3390/molecules28020739_

Round 1

Reviewer 1 Report

The manuscript entitled"Synthesis and Molecular Docking Study of Novel Pyrimidine Derivatives against COVID-19" has been carefully reviewed. In this manuscript, the authors would like to report the synthesis of a novel series of pyrido[2,3-d]pyrimidines; pyrido[3,2-e][1,3,4]triazolo; and tetrazolo[1,5- c]pyrimidines. Then use Auto Dock Vina software to perform molecular docking for the designated compound. The antiviral activity against SARS CoV-2 showed that the tested compounds 7c, 7d and 7e were the most promising antiviral activity compared with the "commonly used protease inhibitor" lopinavir. Although the innovation of the manuscript meets the requirements of the magazine, the manuscript is very rough. Therefore, if the author is willing to revise the manuscript thoroughly, the current reviewer is willing to recommend the manuscript for publication in this journal.

1. The use of some singular and plural forms in the article is wrong, please revise throughout.

2. Figures should align with text.

3. The font of the figures’ title is inconsistent.

4. Scheme 1, the writing format of centigrade is incorrect.

5. In Fig 1, Etzolate” should be revised to “Etazolate

6. The "a,b,c…", "S,R" and "N" in the structure name should be italicized, for example, "[2,3-d]" should be changed to "[2,3-d]", Please check the whole article.

7. In scheme 1, 0C” should be revised to “oC” ; “O2NC6H4” should be revised to “NO2C6H4”, Please check the whole article.

8. In scheme 2 and 5, 3b” should be revised to “3a” ;

9. In scheme 2 and 3, "BrC6H4", the position of "Br" should be indicated.

10. The number of the compound needs to be bold, for example, in line 132 , “compounds 11-14” should be revised to “compounds 11-14” , Please check the whole article.

11. Scheme 1, 2, 5, “4-O2NC6H4” should be revised to “4-NO2C6H4”.

12. “DMSO-d6”, d6 should be italicized.

13. Scheme 4, “:B” should be revised to “BH3”.

14. Scheme 5, “3b” was inconsistent with the text (line 127-133).

15. Line 225, “2a” should be revised to “(2a)”, please revise throughout.

16. Line 340-349, the font of “General procedure for the Synthesis of (8c,d) and (10c,d)” and the number of compounds should be bold.

17. The format of references is inconsistent. For example, the journal names of references 6, 8 and 9 should be abbreviated and italicized. Please check all references.

Author Response

Dear MDPI editorial team,

Thank you for giving me an opportunity to revise my manuscript. We have revised and did the required information as per your suggestions.

The changes made this time have been highlighted in blue color.

Reviewer 2 Report

The manuscript by Al-Amshany et al contains interesting chemistry on the use of heterocyclic systems to obtain biological activity, but the approach taken is outdated.

The authors follow a traditional scheme in which, in the introduction of the manuscript, describe that there are various types of heterocyclic systems with biological activity and then use brute force to obtain a whole series of compounds without any type of prior design of the structures or the substituents to be selected.

Once the compounds are obtained, they carry out a biological study, even with interesting results, and only later do they use computational chemistry to try to justify said results.

It is curious that having in hand all the necessary tools to carry out a modern Medicinal Chemistry approach, that is the computational tools, the synthetic and spectroscopic capabilities, and the biological testing, they still opt for an old-fashioned approach.

The logical sequence today should have been to focus on a type of activity, such as against COVID, and later, once the pocket has been established in the receptor where these molecules must interact, computationally design potentially active structures and select them using docking techniques, and subsequently carry out its synthesis thus increasing the possibilities of obtaining highly biologically active compounds

In addition, having today's tools such as Lipinski’s rule of five to select candidates with greater chances to be orally active or free websites (such as SwissADME, http://www.swissadme.ch/) in which the solubility, log P, and other pharmacokinetic parameters can be predicted, they should have incorporated this type of approach in their proposal. Druggability is important during such a kind of project.

Finally, the text contains some errors in English (such as concordance of verbs and use of connectors) that are normal for those of us who are not native speakers of English, and, in this sense, I would like to recommend the authors to review the text using the Grammarly tool (https://grammarly.com/), which is free and can be used inside Word.

For all of the above and without disregarding the amount of work carried out by the authors, I cannot recommend the publication of the said manuscript in Molecules, since it presents little interest within the field of modern computationally supported Medicinal Chemistry.

Author Response

(The authors gave the same response as above.)

Round 2

Reviewer 2 Report

The authors have improved the manuscript so no reason for not accepting it.

Author Response

Dear MDPI editorial team,

Happy New Year 2023!

Thank you for giving me an opportunity to revise my manuscript. We have revised and did the required information as per your suggestions.

The changes made this time have been highlighted in red color.

See my file attachment

Best Regards
